# Financial Vision-Based Reinforcement Learning Trading Strategy

Yun-Cheng Tsai [1,*], Fu-Min Szu [2], Jun-Hao Chen [3] and Samuel Yen-Chi Chen [4]

1 Department of Technology Application and Human Resource Development, National Taiwan Normal University, 162, Section 1, Heping E. Rd., Taipei City 106209, Taiwan
2 Department of Electrical Engineering, National Taiwan University, 1, Section 4, Roosevelt Rd., Taipei City 106216, Taiwan
3 Department of Computer Science and Information Engineering, National Taiwan University, 1, Section 4, Roosevelt Rd., Taipei City 106216, Taiwan
4 Computational Science Initiative, Brookhaven National Laboratory, 98 Rochester St., Upton, NY 11973, USA
* Correspondence: pecu@ntnu.edu.tw

**Abstract:** Recent advances in artificial intelligence (AI) for quantitative trading have led to its general superhuman performance among notable trading performance results. However, if we use AI without proper supervision, it can lead to wrong choices and huge losses. Therefore, we need to ask why AI makes decisions and how AI makes decisions so that people can trust AI. By understanding the decision process, people can make error corrections, so the need for explainability highlights the artificial intelligence challenges that intelligent technology can explain in trading. This research focuses on financial vision, an explainable approach, and the link to its programmatic implementation. We hope our paper can refer to superhuman performance and the reasons for decisions in trading systems.

**Keywords:** financial vision; Gramian angular field (GAF); reinforcement deep reinforcement learning (Deep RL); proximal policy optimization (PPO)

## 1. Introduction

Suppose that investors want to predict the future transaction price or ups and downs directly. In that case, the fatal assumption is that the training data set is consistent with the data distribution that has not occurred in the future. However, the natural world will not let us know whether the subsequent data distribution will change. Because of this, even if researchers add a moving window to the training process, it is inevitable that "machine learning obstacles-prediction delay" will occur. Our method can avoid "machine learning obstacles-prediction delay". We also propose auto trading by deep reinforcement learning. Our new article makes the following contributions:

1. Our first contribution is not to make future predictions but to focus on the current "candlesticks pattern detection", such as engulfing pattern, morning star, etc.
2. Our second contribution focuses on detecting trading entry and exit signals combined with related investment strategies.
3. Our third contribution is found from experiments that the 15 min price data of the Ethereum train through transfer learning is suitable for US stock trading.

With the rise in deep learning and reinforcement learning technology, breakthrough innovations are in computer trading software [1,2]. Artificial intelligence (AI) is more efficient than a calculation model that only uses static data [3]. Investment companies have used computer algorithms to process transactions for several years. Both new and old investment companies have begun to use artificial intelligence to help customers process investments [4].

Recent advances in AI for quantitative trading have led to its widespread demonstration of superhuman performance in significant trading performance [5]. Likewise,

reinforcement learning (RL) [6] has found tremendous success and demonstrated a superhuman level of capability in the trading field [7].

AI trading starts with supervised learning. The machine can judge the desired result based on the given characteristics through data and specific tags. However, in the era of big data, many data are often not so complete. Many people cannot distinguish between new AI transactions and traditional programming transactions. Organizing each data into a specific format is impossible, so using supervised learning-related algorithms for various situational prediction analyses is impossible. Instead, AI trading must be based on unsupervised learning algorithms, such as deep learning and reinforcement learning. Deep learning has broken through the limitations of previous calculations due to advances in computer science and AI algorithms. Through deep learning and reinforcement learning, hundreds of high-level neural networks that simulate the human brain are calculated.

Some AI computing mechanisms are complex and challenging to understand. However, the potential risk of AI trading is a "black box" decision (see [8,9]). If we use AI without proper supervision, AI may lead to wrong choices and huge losses. Hence, we need to ask about the AI "black box": Why did AI decide to do this or not? Why can people trust AI or not? How can people fix their mistakes? These problems also highlight the challenges that AI technology can explain in trading.

Humans can judge the candlestick pattern by instinct and determine the trading strategies with the corresponding candlestick pattern in the financial market. Could a computer think as humans see? Our goal is to teach machines to understand what they see like humans. For example, the device recognizes candlestick patterns, inferring their geometry, and understanding the market's relationships, actions, and intentions. In addition, we would like to know why the decisions are what AI models are. One must seek explanations of the process behind a model's development, and not just explanations of the model itself.

Our study designs an innovative explainable AI trading framework by combining financial vision with deep reinforcement learning. Hence, we propose a financial vision field, which can understand a candle's critical components and what they indicate to apply candlestick chart analysis to a trading strategy. We combine deep reinforcement learning to realize intuitive trading based on a financial vision to surveillance candlestick. We involve observing a large number of the candlestick, forming automatic responses to various pattern recognition.

With these extraordinary capabilities in automatic control, it is natural to consider RL techniques in algorithmic trading. Indeed, several works have tried to apply RL to trade financial assets automatically.

One of the challenges to constructing an effective RL-based algorithmic trading system is to properly encode the input signal for the agent to make decisions. With the recent advances in convolutional neural networks (CNN), a potential scheme encodes financial time series into images.

This work proposes an algorithmic trading framework based on deep reinforcement learning and the GAF encoding method. Our contributions are the following:

- Provide an algorithmic trading framework for the study of RL-based strategies.
- Demonstrate successful RL trading agents with GAF encoded inputs of price data and technical indicators.

The difference between AI trading is that AI is a target environment (such as the Standard and Poor's 500). AI trading uses unsupervised machine competition to learn. Machine intelligence determines when to place an order or stop selling at a profit. The breakthrough of machine learning in AI trading is to use a new type of unsupervised learning to formulate strategies through data identification features. For example, the golden cross is a multi-feature that allows AI to backtest and learn through over 400 million transaction records in 20 years. As a result, AI robots can find high-profit making money models and gain instant advantages through high-frequency calculations.

The global FinTech industry is already the next economic driving force for investment in many countries. Robo-advisors are already common in advanced European and

American countries. We believe that ordinary investors will gradually accept that financial advisors no longer rely only on high-end franchised services. Wealthy members generally serve the investing public with financial needs. The development of AI-to-AI transactions will enable the financial industry to have a brighter future in upgrading the international AI transaction industry. The sector will fully upgrade to bring users a new investment and financial management experience, creating unprecedented synergies.

Reinforcement learning can interact with the environment and is suitable for applications in decision control systems. Therefore, we used the reinforcement learning method to establish a trading strategy for cryptocurrency and U.S.A. stock markets, avoiding the longstanding unstable trends in profound learning predictions. We found from experiments that the 15-min price data of Ethereum were training through transfer learning after learning the candlesticks pattern suitable for entering and exiting the market of U.S. stock trading. The experimental results demonstrate superior performance compared to the top 10 most popular ETFs. This study focuses on financial vision, explainable methods, and links to their programming implementations. We hope that our paper will serve as a reference for superhuman performance and explain the decisions in the trading system.

The paper is organized as follows. In Section 2, we introduce the concepts of financial vision for candlesticks pattern recognition. In Section 3, we introduce the RL background knowledge used in this work. In Section 4, we describe the proposed GAF-RL trading framework. In Section 5, we describe the experimental procedures and results in detail. Finally we discuss the results in Section 6 and conclude in Section 7.

## 2. Financial Vision

Technical analysis is a specific product through historical data to make trading decisions or signal a general designation. The data include price, volume, and price to obtain the assets on the market reaction. We try to find out the direction of the future changes. This idea is from the people's behavior in the market, which has reproducibility.

The trading account's psychological decision making is from a large proportion of investors. By researching the past, others' trading behavior and believing that this behavior may appear again based on experience, a rational choice can be made.

### 2.1. Candlesticks Pattern

Following the chart is drawn from historical prices according to specific rules. These features help traders to see the price trend. The three more common charts are histograms, line charts, and the most widely used candlestick.

The candlestick originated is from Japan in the 17th century. It has been popular in Europe and the United States for over a century, especially in the foreign exchange market. As the most popular chart in technical analysis, traders should understand it. It is named after a candle, as shown in Figure 1.

Each bar of the candlestick draws from open price, high price, low price, and close price as follows:

1. Open price: this price is the first price that occurs during the period;
2. High price: the highest price that occurs during the period;
3. Low price: the lowest price that occurs during the period;
4. Close price: the last price that occurs during the period.

If the close price is higher than the open price, the candlestick is as follows:

1. The top of the candle body is the close price;
2. The bottom is the open price;
3. The color is usually green or white.

If the close price is lower than the open price, the candlestick is as follows:

1.  The open price above the candle body;
2.  The close price below;
3.  The color is usually red or black.

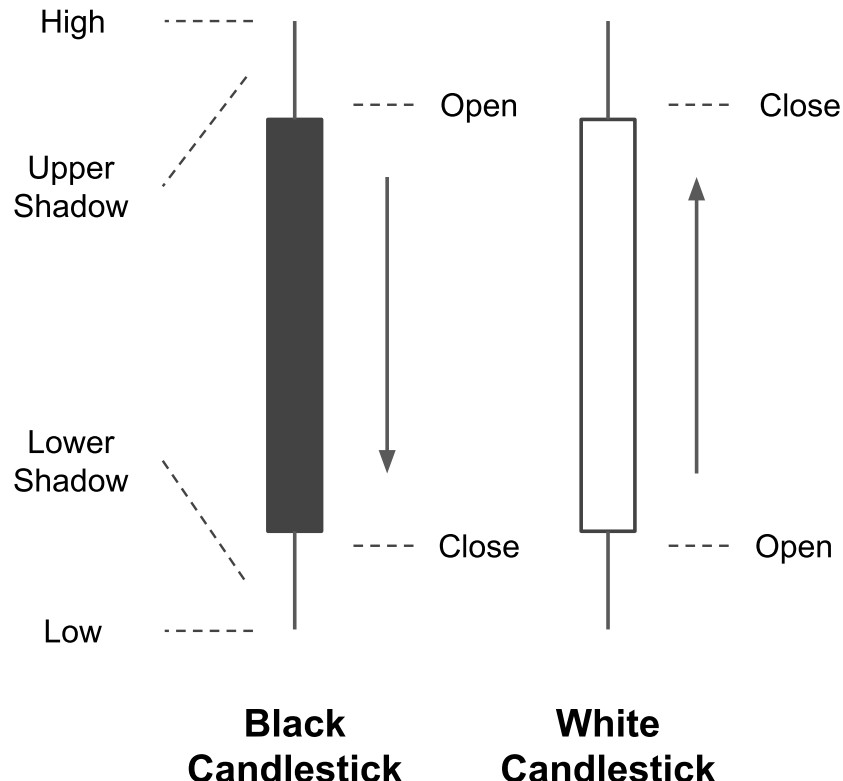

**Figure 1.** The shape of a candlestick.

The candlestick has no hatching in some cases because the open or close price coincides with the high or low price. For example, if the candle is very short, the open and close prices of the candlestick are very similar.

Focusing on the relationship between price and volume, money is the fundamental element in pushing prices, even though the enormous volumes of goods do not necessarily lead to immediate price changes. However, the trading amount reflects the commodity on the market by the degree of attention, can also effectively neutralize dilute price volatility caused by the artificial manipulation, and belongs to the steady and lower the risk of loss of judgment strategy, usually accompanied by another strategy target reference.

### 2.2. Gramian Angular Field (GAF) Encoding

Wang and Oates first proposed GAF as a new framework for encoding time series into images [10]. GAF represents time-series data by replacing the typical Cartesian coordinates with the polar coordinate system and converting angles into symmetric matrices. The Gramian angular summation field is a GAF that uses the cosine function. In the GASF matrix, each element is the cosine of the summation of the angles. GASF can be transformed back to time-series data by diagonal elements. Compared to the Cartesian coordinates, the polar coordinates preserve absolute temporal relations.

There are three steps to encoding time-series data with GAF, shown in Figure 2 [11]:

Firstly, the time-series $X = \{x_1, x_2 \cdots x_n\}$ to be encoded is scaled into the interval $[0, 1]$ via the minimum–maximum scaling in Equation (1).

$$\widetilde{x}_i = \frac{x_i - \min(X)}{\max(X) - \min(X)} \tag{1}$$

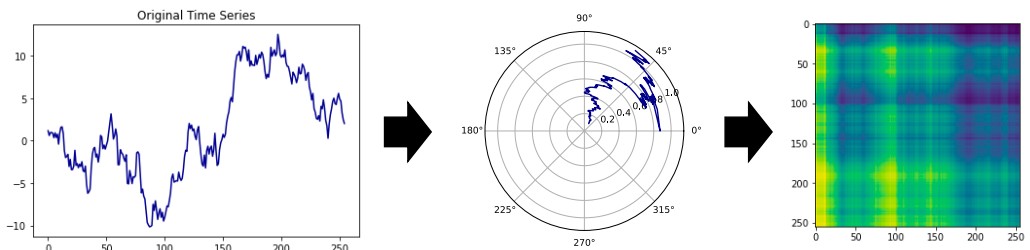

**Figure 2.** The GASF mechanism.

The notation $\widetilde{x}_i$ represents each normalized element from the entire normalized set $\widetilde{X}$. The arc cosine values $\phi_i$ of each $\widetilde{x}_i$ are calculated.

$$\phi_i = \arccos(\widetilde{x}_i), 0 \leq \widetilde{x}_i \leq 1, \widetilde{x}_i \in \widetilde{X} \tag{2}$$

These $\phi$ angles used to generate the GAF matrix are as follows:

$$\begin{aligned}
\text{GAF} &= \cos(\phi_i + \phi_j) \\
&= \begin{bmatrix}
\cos(\phi_1 + \phi_1) & \cdots & \cos(\phi_1 + \phi_n) \\
\cos(\phi_2 + \phi_1) & \cdots & \cos(\phi_2 + \phi_n) \\
\vdots & \ddots & \vdots \\
\cos(\phi_n + \phi_1) & \cdots & \cos(\phi_1 + \phi_n)
\end{bmatrix}
\end{aligned} \tag{3}$$

In this work, we employ the Gramian angular field (GAF) [12] method to encode the time series into images. We choose GAF to encode the candlestick as an image because the candlestick contains time series data. The raw data are hard to find essential features in CNN models [11]. Then we choose GAF-CNN instead of the CNN model because the candlestick charts have time series data. It is hard to find essential features in CNN models. However, the GAF-CNN model finds the parts easily. We take the morning star pattern as an example (Figure 3). There is a downtrend, and the trend's end is the pattern of two long candlesticks sandwiching a short one. Figure 4 is the features found in the second conversion layer of the CNN model. Figure 5 is the features found in the second conversion layer of the GAF-CNN model. In the CNN model, the downtrend is not here, and some outputs have no attributes. The GAF-CNN model finds some particular patterns. Hence, we choose the GAF-CNN model. After introducing the GAF-CNN model, we introduce some methods that can use the model to define the training data.

The generated GAF $n \times n$ matrix, where $n$ is the length of sequence considered, is used as input to the CNN. This method allows it to keep the temporal information while avoiding recurrent neural networks, which are computationally intensive. GAF encoding methods are employed in various financial time-series problems [11–13].

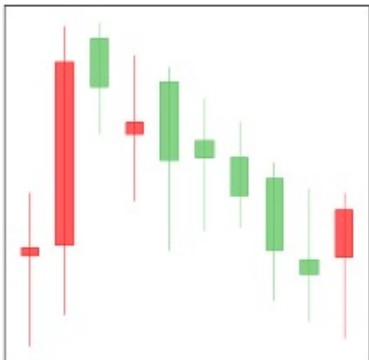

**Figure 3.** An example of morning star pattern.

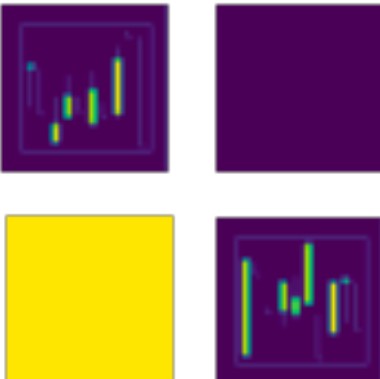

**Figure 4.** CNN output.

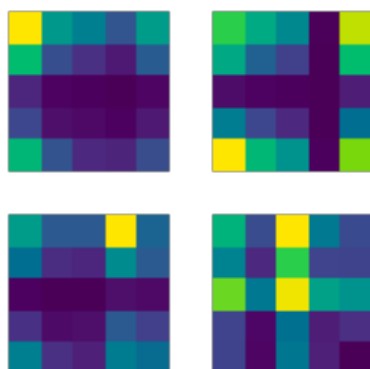

**Figure 5.** GAF-CNN output.

## 3. Reinforcement Learning

Reinforcement learning (RL) is a machine learning paradigm in which an *agent* learns how to make decisions via interacting with the environments [6]. The reinforcement learning model comprises an agent. The agent performs an action based on the current state. The action receives from the environment and returns feedback to the agent. The input can be either a reward or a penalty. Once the agent receives the reward, they adjust the relative function between the state and the action to maximize the expected return. The function could be a value function or a policy function.

A value function refers to the reward from a particular action in a specific state. Therefore, accurately estimating the value function is essential to the model. Conversely, underestimating or overestimating the value of certain conditions or actions would influence the learning performance. A policy function is ideal for achieving a maximum expected return in a particular state. In the reinforcement learning model, actions that

maximize the expected return (value) in a specific condition are called policies. In several advanced models, policy functions directly apply to maximize expected returns.

Concretely speaking, the *agent* interacts with an *environment* $\mathcal{E}$ over a number of discrete time steps. At each time step $t$, the agent receives a *state* or *observation* $s_t$ from the environment $\mathcal{E}$ and then chooses an *action* $a_t$ from a set of possible actions $\mathcal{A}$ according to its *policy* $\pi$. The policy $\pi$ is a function which maps the state or observation $s_t$ to action $a_t$. In general, the policy can be stochastic, meaning that given a state $s$, the action output can be a probability distribution $\pi(a_t|s_t)$ conditioned on $s_t$. After executing the action $a_t$, the agent receives the state of the next time step $s_{t+1}$ and a scalar *reward* $r_t$. The process continues until the agent reaches the terminal state or a pre-defined stopping criterion (e.g., the maximum steps allowed). An *episode* is defined as an agent starting from a randomly selected initial state and following the aforementioned process all the way through the terminal state or reaching a stopping criteria.

We define the total discounted return from time step $t$ as $R_t = \sum_{t'=t}^{T} \gamma^{t'-t} r_{t'}$, where $\gamma$ is the discount factor that lies in $(0, 1]$. In principle, $\gamma$ is from the investigator to control how future rewards weigh the decision-making function. When we use a large $\gamma$, the agent weighs the future reward more heavily. On the other hand, future rewards are ignored quickly with a small $\gamma$. The immediate rewards will weigh more. The goal of the agent is to maximize the expected return from each state $s_t$ in the training process. The *action-value function* or *Q-value function* $Q^\pi(s, a) = \mathbb{E}[R_t|s_t = s, a]$ is the expected return for selecting an action $a$ in state $s$ based on policy $\pi$. The optimal action value function $Q^*(s, a) = \max_\pi Q^\pi(s, a)$ gives a maximal action-value across all possible policies. The value of state $s$ under policy $\pi$, $V^\pi(s) = \mathbb{E}[R_t|s_t = s]$ is the agent's expected return by following policy $\pi$ from the state $s$. Various RL algorithms are designed to find the policy which can maximize the value function. The RL algorithms which maximize the value function are called *value-based* RL.

### 3.1. Policy Gradient

In contrast to the value-based RL, which learns the value function and use it as the reference to generate the decision on each time-step, there is another kind of RL method called the *policy gradient*. In this method, the policy function $\pi(a|s; \theta)$ is parameterized with the parameters $\theta$. The $\theta$ is then subject to the optimization procedure, which is the *gradient ascent* on the expected total return $\mathbb{E}[R_t]$. One of the classic examples of the policy gradient algorithm is the REINFORCE algorithm [14]. In the standard REINFORCE algorithm, the parameters $\theta$ are updated along the direction $\nabla_\theta \log \pi(a_t|s_t; \theta) R_t$, which is the unbiased estimate of $\nabla_\theta \mathbb{E}[R_t]$. However, the policy gradient method suffers from large variance of the $\nabla_\theta \mathbb{E}[R_t]$, making the training very hard. To reduce the variance of this estimate and keep it unbiased, one can subtract a learned function of the state $b_t(s_t)$, which is known as the *baseline*, from the return. The result is, therefore, $\nabla_\theta \log \pi(a_t|s_t; \theta)(R_t - b_t(s_t))$.

### 3.2. Advantage Actor-Critic (A2C)

A learned estimate of the value function is a common choice for the baseline $b_t(s_t) \approx V^\pi(s_t)$. This choice usually leads to a much lower variance estimate of the policy gradient. When one uses the approximate value function as the baseline, the quantity $R_t - b_t = Q(s_t, a_t) - V(s_t)$ can be seen as the *advantage* $A(s_t, a_t)$ of the action $a_t$ at the state $s_t$. Intuitively, one can see this advantage as "how good or bad the action $a_t$ compared to the average value at this state $V(s_t)$." For example, if the $Q(s_t, a_t)$ equals 10 at a given time step $t$, it is not clear whether $a_t$ is a good action or not. However, if we also know that the $V(s_t)$ equals, say, 2 here, then we can imply that $a_t$ may not be bad. Conversely, if the $V(s_t)$ equals 15, then the advantage is $10 - 15 = -5$, meaning that the $Q$ value for this action $a_t$ is well below the average $V(s_t)$ and therefore that action is not good. This approach is called the *advantage actor–critic* (A2C) method, where the policy $\pi$ is the actor and the baseline, where the value function $V$ is the critic [6].

### 3.3. Proximal Policy Optimization (PPO)

In the policy gradients method, we optimize the policy according to the *policy loss*

$$L_{\text{policy}}(\theta) = \mathbb{E}_t[-\log \pi(a_t \mid st; \theta)]$$

via gradient descent.

However, the training itself may suffer from instabilities. If the step size of the policy update is too small, the training process would be too slow. On the other hand, if the step size is too large, there will be a high variance in training. The proximal policy optimization (PPO) [15] fixes this problem by limiting the policy update step size at each training step. The PPO introduces the loss function called the *clipped surrogate loss function* that will constrain the policy change to a small range with the help of a clip. Consider the ratio between the probability of action $a_t$ under the current policy and the likelihood under the previous policy

$$q_t(\theta) = \frac{\pi(a_t \mid s_t; \theta)}{\pi(a_t \mid s_t; \theta_{\text{old}})}.$$

If $q_t(\theta) > 1$, it means the action $a_t$ is with a higher probability in the current policy than in the old one. Additionally, if $0 < q_t(\theta) < 1$, it means that the action $a_t$ is less probable in the current policy than in the old one. Our new loss function can then be defined as

$$L_{\text{policy}}(\theta) = \mathbb{E}_t[q_t(\theta)A_t] = \mathbb{E}_t[\frac{\pi(a_t \mid s_t; \theta)}{\pi(a_t \mid s_t; \theta_{\text{old}})}A_t],$$

where

$$A_t = R_t - V(s_t|\theta)$$

is the advantage function.

If the action under the current policy is much more probable than in the previous approach, the ratio $q_t$ may be significant, leading to a considerable policy update step. The original PPO algorithm [15] circumvents this problem by adding a constraint on the ratio, which can only be in the range 0.8 to 1.2. The modified loss function is now

$$L_{\text{policy}}(\theta) = \mathbb{E}_t[-min(q_t A_t, clip(q_t, 1 - C, 1 + C)A_t)],$$

where the $C$ is the clip hyperparameter (common choice is 0.2). Finally, the value loss and entropy bonus add to the total loss function as usual:

$$L(\theta) = L_{\text{policy}} + c_1 L_{\text{value}} - c_2 H,$$

where

$$L_{\text{value}} = \mathbb{E}_t[\|R_t - V(s_t|\theta)\|^2]$$

is the value loss and $H = \mathbb{E}_t[H_t] = \mathbb{E}_t[-\sum_j \pi(a_j \mid s_t; \theta) \log(\pi(a_j \mid s_t; \theta))]$ is the entropy bonus, which is to encourage exploration.

### 3.4. PPO Algorithms

PPO allows us to train AI policies in challenging environments. With supervised learning, we can quickly implement a cost function, run gradient descent, and be confident that we will get great results with relatively little hyperparameter tuning. According to the original PPO algorithm [15], the Algorithm 1 is the pseudocode of the PPO training algorithm.

---

**Algorithm 1** PPO algorithmic

---

Define the number of total episode $M$
Define the maximum steps in a single episode $S$
Define the update timestep $U$
Define the update epoch number $K$
Define the epsilon clip $C$
Initialize trajectory buffer $\mathcal{T}$
Initialize time step counter $t$
Initialize two sets of model parameters $\theta$ and $\theta_{\text{old}}$
**for** episode $= 1, 2, \ldots, M$ **do**
    Reset the testing environment and initialize state $s_1$
    **for** step $= 1, 2, \ldots, S$ **do**
        Update the time step $t = t + 1$
        Select the action $a_t$ from the policy $\pi(a_t \mid s_t; \theta_{\text{old}})$
        Execute action $a_t$ in emulator and observe reward $r_t$ and next state $s_{t+1}$
        Record the transition $(s_t, a_t, \log \pi(a_t \mid s_t; \theta_{\text{old}}), r_t)$ in $\mathcal{T}$
        **if** $t = U$ **then**
            Calculate the discounted rewards $R_t$ for each state $s_t$ in the trajectory buffer $\mathcal{T}$
            **for** $k = 1, 2, \ldots, K$ **do**
            Calculate the log probability $\log \pi(a_t \mid s_t; \theta)$, state values $V(s_t, \theta)$ and entropy $H_t$.
            Calculate the ratio $q_t = \exp\left(\log \pi(a_t \mid s_t; \theta) - \log \pi(a_t \mid s_t; \theta_{\text{old}})\right)$
            Calculate the advantage $A_t = R_t - V(s_t, \theta)$
            Calculate the $surr_1 = q_t \times A_t$
            Calculate the $surr_2 = clip(q_t, 1 - C, 1 + C) \times A_t$
            Calculate the loss $L = \mathbb{E}_t[-min(surr_1, surr_2) + 0.5\|V(s_t, \theta) - R_t\|^2 - 0.01 H_t]$
            Update the agent policy parameters $\theta$ with gradient descent on the loss $L$
            **end for**
            Update the $\theta_{\text{old}}$ to $\theta$
            Reset the trajectory buffer $\mathcal{T}$
            Reset the time step counter $t = 0$
        **end if**
    **end for**
**end for**

---

## 4. Financial Vision Based RL Trading Framework

Figure 6 is the framework of Algorithm 2. There are two large architectures, one of which is part of GAF encoding (Algorithm 3), and the other is the actor–critic PPO architecture. GAF encoding aims to analyze time sequences in a way that is different from traditional time sequences. To extract the time sequence characteristics from another perspective, we must independently train pattern recognition before the experiment to make its accuracy rise to a certain level.

For example, take the probability distribution of patterns and the input model of reinforcement learning. This study used the ETH/USD exchange data for 15 min from 1 January 2020 to 1 July 2020 as a reference for the system's environment design and performance calculation. The period is the training period for the agent. The agent repeatedly applied the data to learn in this period, eventually obtaining an optimal investment strategy.

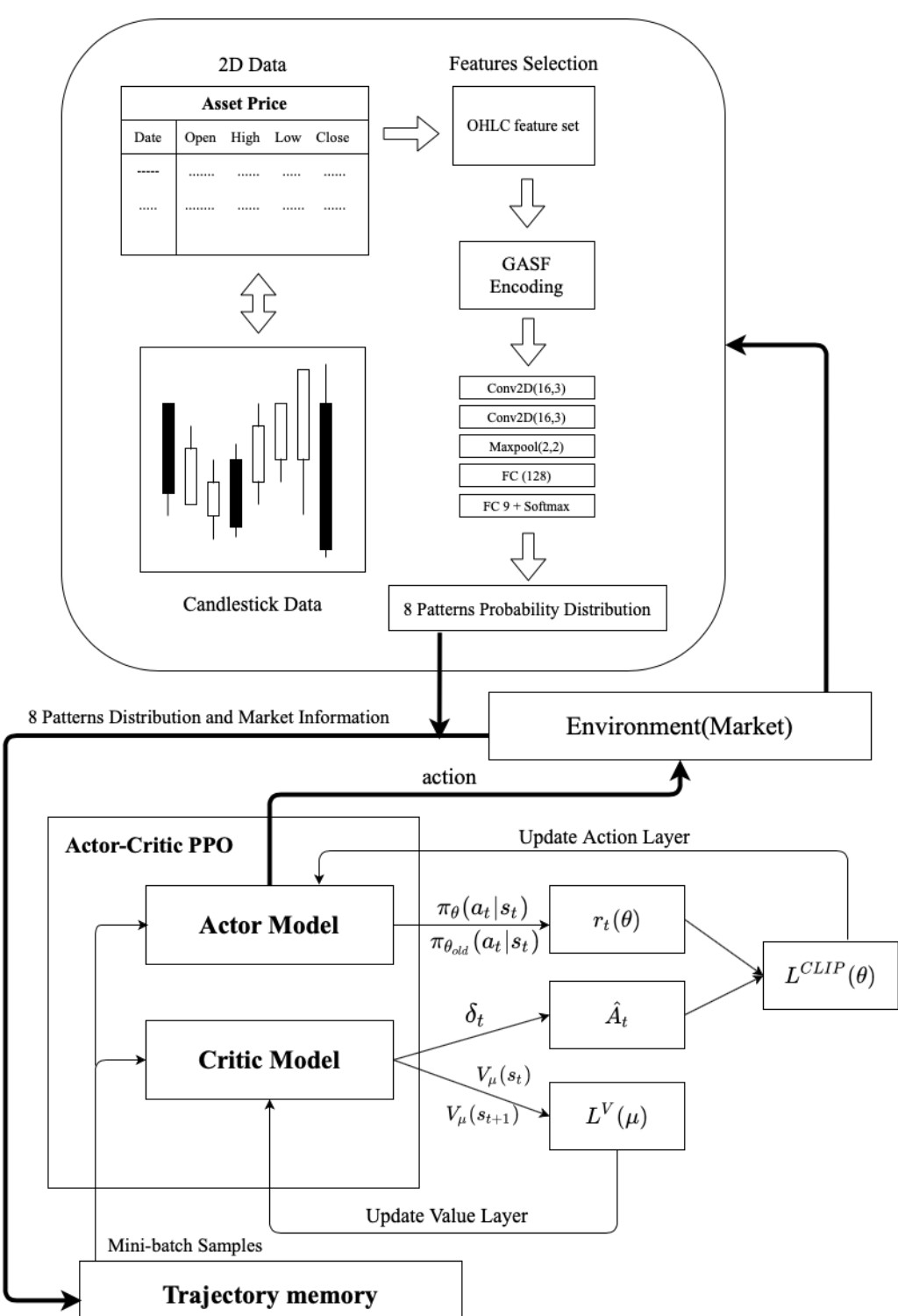

**Figure 6.** GAF-RL framework.

---

**Algorithm 2** PPO for GAF-based algorithmic trading

---

Define the state of probability distribution and max timestep from pattern recognition $P, T_{max}$

Define the environment state with the probability distribution $s = P$

Define the number of total episode $M$

Define the maximum steps in a single episode with max timestep $S = T_{max}$

Define the update time step $U$

Define the update epoch number $K$

Define the epsilon clip $C$

Initialize trajectory buffer $\mathcal{T}$

Initialize time step counter $t$

Initialize two sets of model parameters $\theta$ and $\theta_{\mathrm{old}}$

**for** episode $= 1, 2, \ldots, M$ **do**

    Reset the testing environment and initialize state $s_1 = P_1$

    **for** step $= 1, 2, \ldots, S$ **do**

        Update the time step $t = t + 1$

        Select the action $a_t$ from the policy $\pi(a_t \mid s_t; \theta_{\mathrm{old}})$

        Execute action $a_t$ in emulator and observe reward $r_t$ and next state $s_{t+1} = P_{t+1}$

        Record the transition $(s_t, a_t, \log \pi(a_t \mid s_t; \theta_{\mathrm{old}}), r_t)$ in $\mathcal{T}$

        **if** $t = U$ **then**

            Calculate the discounted rewards $R_t$ for each state $s_t$ in the trajectory buffer $\mathcal{T}$

            **for** $k = 1, 2, \ldots, K$ **do**

            Calculate the log probability $\log \pi(a_t \mid s_t; \theta)$, state values $V(s_t, \theta)$ and entropy $H_t$.

            Calculate the ratio $q_t = \exp\left(\log \pi(a_t \mid s_t; \theta) - \log \pi(a_t \mid s_t; \theta_{\mathrm{old}})\right)$

            Calculate the advantage $A_t = R_t - V(s_t, \theta)$

            Calculate the $surr_1 = q_t \times A_t$

            Calculate the $surr_2 = clip(q_t, 1 - C, 1 + C) \times A_t$

            Calculate the loss $L = \mathbb{E}_t[-min(surr_1, surr_2) + 0.5\|V(s_t, \theta) - R_t\|^2 - 0.01H_t]$

            Update the agent policy parameters $\theta$ with gradient descent on the loss $L$

            **end for**

            Update the $\theta_{\mathrm{old}}$ to $\theta$

            Reset the trajectory buffer $\mathcal{T}$

            Reset the time step counter $t = 0$

        **end if**

    **end for**

**end for**

---

**Algorithm 3** Pattern recognition scheme based on GAF and CNN

---

Define the OHLC of asset with trade date $A$

Define the window sizes of trade date $W$

Define the total length of trade date $T_{total}$

Define the function GAF encoding $GAF(x)$

Define the function CNN model $CNN(x)$

Define the PPO algorithm $PPO$

Define the max time step $T_{max} = T_{total} - W$

**for** $t = 1, 2, \ldots, T_{max}$ **do**

    Calculate the asset OHLC to GAF encoding $A_{gaf} = GAF(A[t : t + W])$

    Calculate the output layer from CNN and softmax layer $CNN(A_{gaf})$

    Define the probability distribution of eight patterns $P_t = CNN(A_{gaf})$

    Train the $PPO$ model by environment state $P_t$

**end for**

---

However, when it comes time to train the PPO's trading strategy, initial market information and Pattens' identification information are brought into the PPO's strategy model. The market information includes technical indicators, bid strategies, and open interest and volume state. The PPO model takes place after a sufficient number of trajectories is collected. It is input into the PPO model of the actor–critic. There are two different layers: one is the action layer, and the other is the value layer. The output layer of the action layer will have three trading signals, namely, buy, sell and hold. This strategy sets a limit of the chips to three times at most. The value layer is to determine the potential value of each state, and the difference with the reward $\delta_t$ to create the advantage function $\hat{A}_t$, $V_\mu(S_t)$ and $V_\mu(S_{t+1})$ is simultaneously generated to update the value layer gradient.

In the PPO training framework, off-policy training is the old policy $\pi_{\theta_{old}}(a_t|s_t)$ interact with the environment produce an important sample to produce another policy gradient to update policy network $\pi_\theta(a_t|s_t)$. After sampling several times, the old policy weights to be updated as the weight of the new policy. The clipped surrogate loss function will constrain the policy change in a small range. To avoid too much divergence between the distribution of the two policies during training, resulting in an unstable training situation. The actions output by the action layer interacts with the following environment to establish the whole training cycle.

## 5. Experiments and Results

*Experiment Data*

Figure 7 shows the training materials from the cryptocurrency ETH/USD for 15 min from 1 January 2020 to 1 July 2020. Due to the characteristics of this kind of commodity, it has the advantages of high volatility, continuous trading time, and significant trading volume. Therefore, there is enough information to learn more effective commodity trading strategies.

The experiment results are from two parts as follows:

1. The first part is the training patterns of GAF-CNN.
2. The second part is the strategies of PPO-RL.

The training patterns are from eight common candlestick patterns. Then the good enough pattern recognition is trained in advance for the second part of the PPO training framework.

The data object of this test is included across time from January 2019 to September 2020. In addition, the outbreak of the COVID-19 pandemic is contained during this period to test whether the stability of this trading strategy is enough to act as an effective hedge tool. In the experiment, four types of commodities in different markets, bull, bear, and stable markets, are selected for comparison. Since many trading platforms do not require fees, this does not consider the transaction fees. When the market volatility is more violent, the return will correlate with growth in terms of profit performance. Conversely, when the market volatility is lessened, the return on profit will reduce.

Figure 8 shows the trading performance of the GAF-PPO model in the cryptocurrency market from August to 2020 June 2021. We want to explain why the commission is working, and the candlestick and the buy and sell points of the GAF-PPO model can give the reasons. Figure 9 shows that the buy and sell points appear on the candlestick specific pattern. For example, the buy points follow the morning star, bullish engulfing, inverted hammer, and bullish Harami patterns. The sell points follow the evening star, bearish engulfing, shooting star, and bearish Harami patterns (the details are in Appendix A).

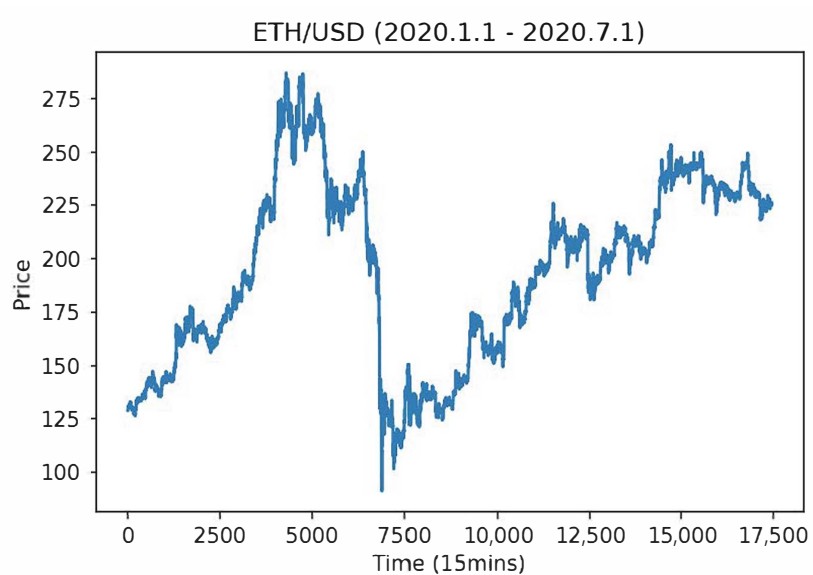

**Figure 7.** ETH/USD 15 min (1 January 2020–1 July 2020).

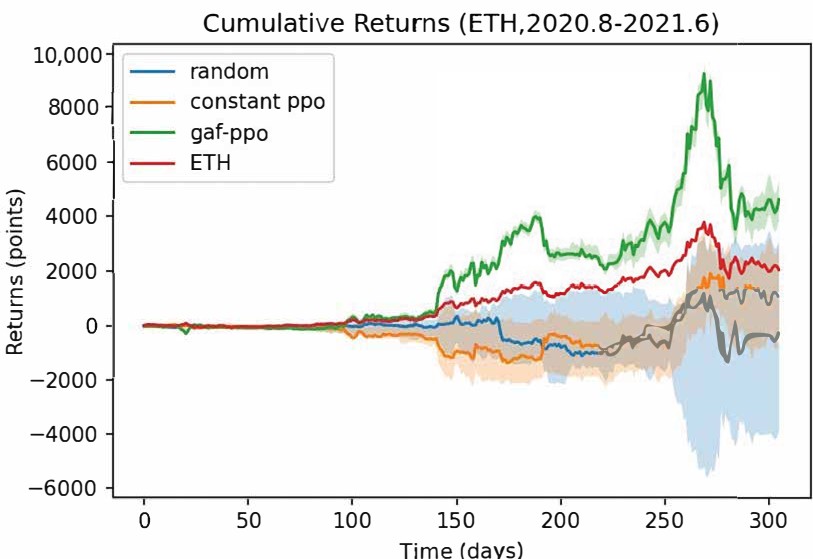

**Figure 8.** The trading performance of GAF-PPO model in the cryptocurrency market.

The attributes of the cryptocurrency market are relatively volatile and easily affected by the news. Therefore, the strategy runs for the cryptocurrency market from August 2020 to June 2021. The market rapidly fluctuates several times due to celebrities supporting and opposing cryptocurrency. Nevertheless, the strategy can still retain profit and a hedging ability in such a market state. The result also proves that this trading strategy has a significant responsibility in the market with severe fluctuations.

The following period is the agent's performance evaluation period. The agent uses a trading strategy to form decisions. The system accumulates the rewards obtained during the evaluation period, which serve as a reference for performance. Figure 10 shows the buy and sell points of the GAF-PPO model in the two large bull markets (Nasdaq: GOOG and NYSE: SPY). Figure 11 shows that the performance of GAF-PPO model has had tremendous growth momentum over the past decade. It has also suffered a certain degree of decline risk after the epidemic outbreak, as seen from the experimental results. During the epidemic, the strategy of the whole model shifts to the conservative and straightforward approach. The density of trades will decrease significantly, but it can also be excellent to catch into

the market to buy in the rapid price pullback. In terms of transaction frequency, there are about two or three transactions a week.

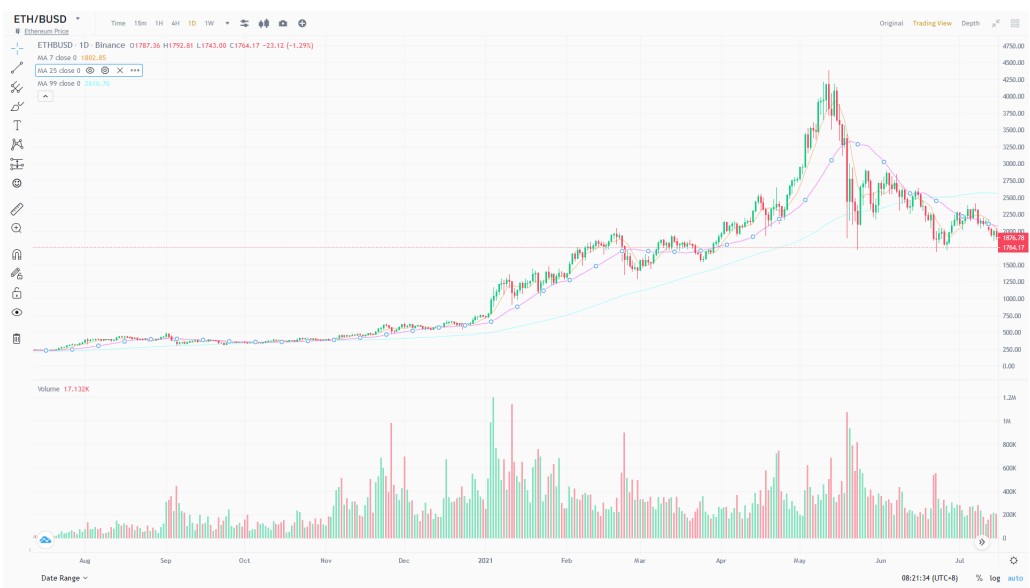

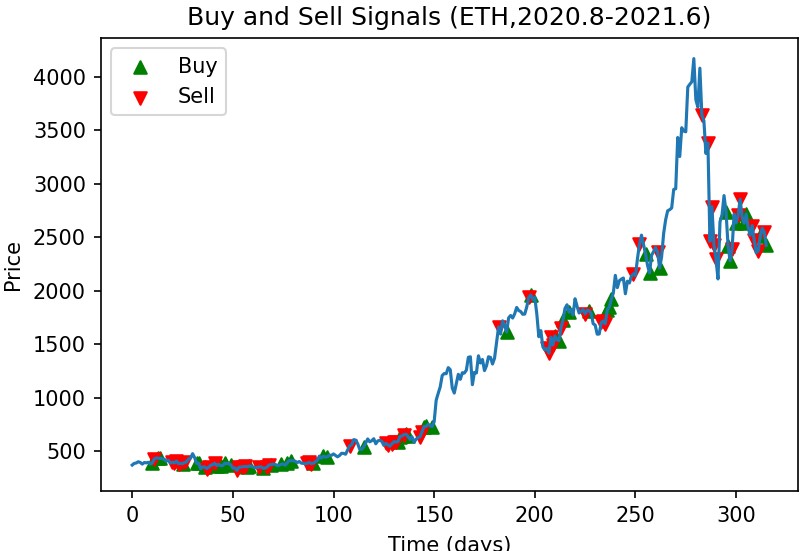

**Figure 9.** The buy and sell points of GAF-PPO model in the cryptocurrency market.

Figure 12 shows the buy and sell points of the GAF-PPO model in the other two assets. It has continued to enter the bear and stable market (USCF: BNO and iShares MSCI Japan ETF: EWJ). Figure 13 shows that the performance of the GAF-PPO model has experienced a considerable decline during this period. The PPO agent turned in the short direction, obtained a significant return, and bought back reasonably. The profit from this short bet accounted for most of the total gain. In the part of EWJ, the initial volatility of the index did not obtain a significant return, representing the inadequacy of the trading strategy for the market with severe fluctuations. Nevertheless, in a word, this trading strategy based on pattern recognition can indeed accurately provide appropriate strategies as hedging operations for investors when downside risks suddenly appear.

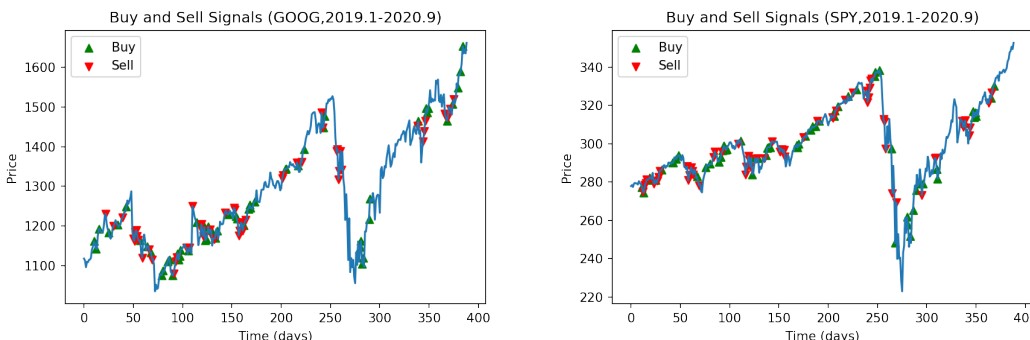

**Figure 10.** The buy and sell points of GAF-PPO model in the bull market during COVID-19 outbreak.

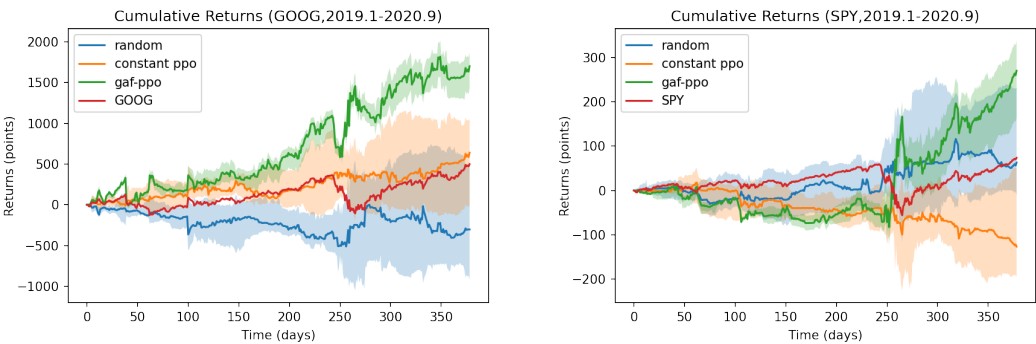

**Figure 11.** The performance of GAF-PPO model in the bull market during COVID-19 outbreak.

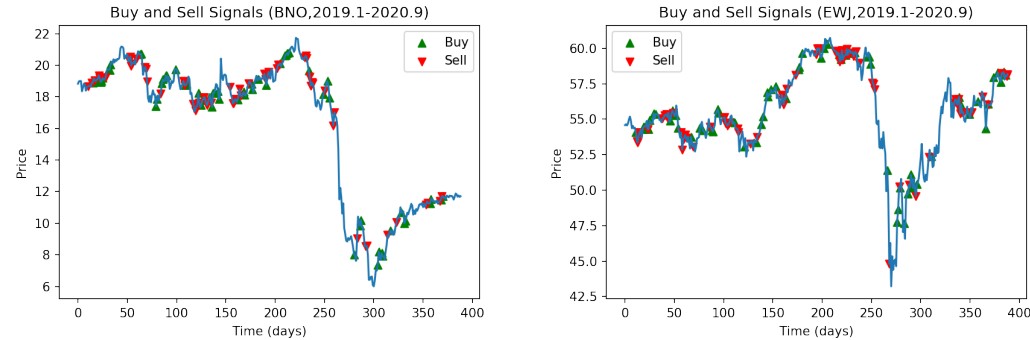

**Figure 12.** The buy and sell points of GAF-PPO model in the bear and rebound market during COVID-19 outbreak.

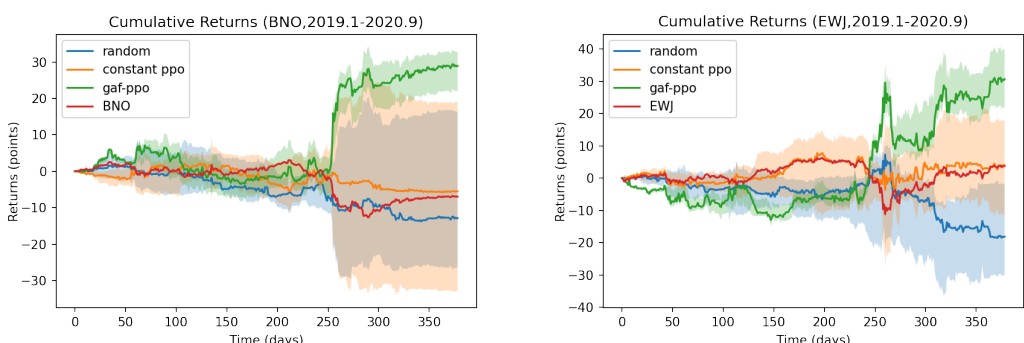

**Figure 13.** The performance of GAF-PPO model in the bear and rebound market during COVID-19 outbreak.

## 6. Discussion

Suppose that license plate recognition is a mature commercial product in computer vision. In that case, our financial vision is the trade surveillance system in financial markets. We want to use the concept that the camera on the road has been monitoring whether there are violations to illustrate the core contribution and use of financial vision in transactions. First, let us describe computer vision technology to monitor violations at intersections, such as the characteristics of speeding, illegal turns, and running red lights. When a camera with license plate recognition capabilities captures the particular patterns, the traffic team will receive a notification and compare the red list with sending the photo to the offender. Using financial vision technology to monitor the price changes of the subject matter is like watching intersections. Investors set specific candlesticks patterns (for example, W bottom, M head, and N-type) that require special attention, and will receive the pattern hunter's notification. Then, how to conduct transactions is another matter. Moreover, how is this monitoring logic different from traditional price prediction?

Traditional price prediction uses the price of the past period $(0, t-1)$ as the input $X$ of supervised learning. The price at time $t$ is the output $Y$ (or directly uses the rise or fall as the learning target). These all imply a fatal assumption: the forecast time point in the real market in the future, the distribution of these data, and the past historical data are consistent. In any case, it divides into a training set and a test set (despite the concept of moving window uses).

The camera only cares about denying the offending vehicle's license plate when discussing intersection surveillance. The camera does not predict whether a car will violate the regulations the next time. Therefore, building the monitoring model will recognize the license plate independently outside of these road conditions. The road condition of this camera at the time $(0, t-1)$ does not regard as input $X$ of supervised learning, and the road condition at time $t$ is regarded as output $Y$. Therefore, the transaction monitoring system constructs a financial vision to learn the candlesticks pattern map. When the candlesticks pattern appears (as if a red light is run), the camera will take a picture and notify the traffic team, list it, and inform the user.

We live in an era in which cryptocurrency is at a crossroads of transitional adaptation that requires more sensitive approaches to adapt to the evolving financial market. By conducting the transfer learning of cryptocurrency trading behavior, the study seeks any possibilities of implementing the trading pattern in the transactions of traditional commodities. With the candlestick analysis, the model attempts to discover feasible rules for establishing a specific practice. Subsequently, deep reinforcement learning utilizes the trading pattern in broadening possible scenarios. We endeavor to comprehend profitability strategies and market sentiment in various conditions to satisfy voluminous risk aversion demands in the market. The experiment results reveal that more accurate judgments can be made when the commodity price fluctuates with volatility. On the contrary, most people tend to be influenced by emotional bias. The inclination acquaints human beings with more stable and long-term markets. Human beings are adapted to be blind-spotted and miscalculate the best entry point when volatile movement occurs. Integrating the machine learning model application can effectively compensate for human deficiency and enhance the efficiency of the trading process. Nevertheless, optimizing the trading frequency to consolidate the liquidity of different financial assets is another issue that deserves more attention in the dynamic financial market.

## 7. Conclusions

Investors want to predict the current wise investment's future transaction price or ups and downs. The fatal assumption is that the training data set is consistent with the data distribution that has not occurred in the future. However, the natural world will not let us know whether the subsequent data distribution will change.

Because of this, even if researchers add a moving window to the training process, it is inevitable that "machine learning obstacles-prediction delay" will occur. Just search

for "machine learning predict stock price", and researchers can find articles full of pits, all of which have this shortcoming. Therefore, our first contribution is not to make future predictions, but to focus on the current "candlesticks pattern detection," such as engulfing pattern, morning star, etc.

However, the "candlesticks pattern" is usually a sensational description. It cannot become a stylized trading strategy if investors cannot write a program to enumerate all the characteristics. Even if a trader has a sense of the market and knows which patterns have to enter and exit, they cannot keep their eyes on all the investment targets. Moreover, our second contribution focuses on detecting trading entry and exit signals combined with related investment strategies.

Finally, we found from experiments that the 15-min price data of the Ethereum train through transfer learning are suitable for US stock trading. The experimental results demonstrate superior performance compared to the top 10 most popular ETFs. This study focuses on financial vision, explainable methods, and links to their programming implementations. We hope that our paper will reference the superhuman performance and explain why the decisions are in the trading system.

**Author Contributions:** Conceptualization, Y.-C.T. and S.Y.-C.C.; methodology, F.-M.S.; software, F.-M.S. and J.-H.C.; validation, Y.-C.T., F.-M.S. and J.-H.C.; formal analysis, Y.-C.T.; investigation, F.-M.S.; resources, Y.-C.T.; data curation, F.-M.S.; writing-original draft preparation, F.-M.S.; writing—review and editing, Y.-C.T.; visualization, F.-M.S.; supervision, Y.-C.T.; project administration, Y.-C.T. All authors have read and agreed to the published version of the manuscript.

**Funding:** This research received no external funding.

**Institutional Review Board Statement:** Not applicable.

**Informed Consent Statement:** Not applicable.

**Data Availability Statement:** The data is from the folder (Deep Reinforcement Learning for Foreign Exchange Trading) of https://github.com/pecu/FinancialVision/.

**Conflicts of Interest:** The authors declare no conflict of interest.

## Appendix A

The eight patterns used in this study are described entirely in this section. The following eight figures illustrate the critical rules that each pattern requires. The white candlestick represents a rising price on the left-hand side of each figure, and the black candlestick represents a dropping price. The arrow indicates the trend. The upward arrow indicates a positive direction, and the downward arrow indicates a negative trend. The text descriptions on the right-hand side are the fundamental rules from the major candlestick signals [16].

### Evening Star Pattern

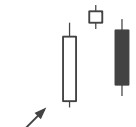

1. The uptrend has been apparent.

2. The body of the first candle is white, continuing the current trend.

3. The second candle should close at least halfway down the white candle.

### Morning Star Pattern

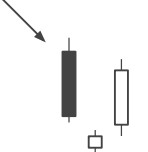

1. The downtrend has been apparent.

2. The body of the first candle is black, continuing the current trend.

3. The third day candle should close at least halfway up the black candle.

### Bearish Engulfing Pattern

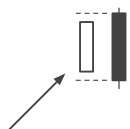

1. The uptrend has been apparent.

2. The body of the first candle is white, and the second is black.

3. The second candle should engulf the first candle with higher open and lower close price.

### Bullish Engulfing Pattern

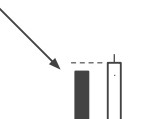

1. The downtrend has been apparent.

2. The body of the first candle is black, and the second is white.

3. The second candle should engulf the first candle with higher close and lower open price.

### Shooting Star Pattern

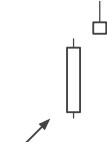

1. The uptrend has been apparent.

2. The body of the first candle is white, continuing the current trend.

3. The body of the second candle is not important, but should be small enough.

4. The lower shadow of second candle should be very small, and the upper shadow should be large enough.

### Inverted Hammer Pattern

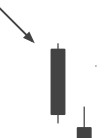

1. The downtrend has been apparent.

2. The body of the first candle is black, continuing the current trend.

3. The body of the second candle is not important, but should be small enough.

4. The lower shadow of second candle should be very small, and the upper shadow should be large enough.

### Bearish Harami Pattern

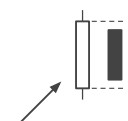

1. The uptrend has been apparent.

2. The body of the first candle is white, and the second is black.

3. The second candle should be engulfed by the first candle with lower open and higher close price.

### Bullish Harami Pattern

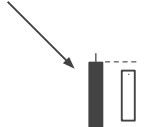

1. The downtrend has been apparent.

2. The body of the first candle is black, and the second is white.

3. The second candle should be engulfed by the first candle with lower close and higher open price.

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
