# Peer review of "Financial Vision-Based Reinforcement Learning Trading Strategy"

_2813-2203, doi:10.3390/analytics1010004_

Round 1
Reviewer 1 Report
The suggestions for the paper:
1. Mathematical modeling is acceptable
2. Include hardware structure
3. Explain more in the proposed algorithm
4. How to fix error criteria and stopping condition
5. Include flowchart for the proposed work.
Reviewer 2 Report
This study proposes a vision-based method for learning candle charts and developing optimal strategies.However, there are a few corrections that need to be addressed.
1. Even though this is your first draft, please write according to the MDPI format.
2. There is insufficient explanation as to why it is necessary to train the candle chart as it is as an image. What is missing in reinforcement learning using only the raw data as-is as training data?
3. The results of the explanation in the black box should be described in detail.
Reviewer 3 Report
Advantages:
Candlestick patterns, which are technical trading tools, have been used for centuries to predict price direction.There are various candlestick patterns used to determine price direction and momentum.
Not all candlestick patterns work equally well. Their huge popularity has lowered reliability because they've been analyzed by hedge funds and their algorithms. These well-funded players rely on lightning-speed execution to trade against retail investors and traditional fund managers who execute technical analysis strategies found in popular texts.
This is a good analytical report with practical experiments implementation. The mathematical part is acceptable.
Main moto of manuscript: “Our study designs an innovative explainable A.I. trading framework…”
The design of manuscript is quality.
Disadvantages:
Some remarks and questions :
- - too many keywords
- - in the introduction part I found repeating text from abstract: “Some AI computing mechanism…”
- - in the introduction part: how you plan to control the correctness of the AI solution for trading?
- - choose one label for artificial intelligence: AI or A.I.
- - in 3.4. what is stopping criteria for PPO?
- - In 4. which will change significantly if the exchange data, for example, is taken in May 26, 2022.?
- - In 4.1. in which step in the algorithm 2 you used PPO for GAF?
- - the algorithm 3 should be commented on more because the end goal is not understood.
